# Early Gestational Diabetes Mellitus Diagnosis: A Strategy for Mitigating Excessive Maternal Weight Gain—LINDA-Brasil Study

**DOI:** 10.3390/nu17162600

**Published:** 2025-08-10

**Authors:** Letícia Ribeiro Pavão da Silveira, Maria Inês Schmidt, Paula Bracco, Rita Mattiello, Michele Drehmer

**Affiliations:** 1Postgraduate Program in Epidemiology, School of Medicine, Universidade Federal do Rio Grande do Sul, Porto Alegre 90035-003, Rio Grande do Sul, Brazil; nutrileticiarps@gmail.com (L.R.P.d.S.); maria.schmidt@ufrgs.br (M.I.S.); paula.abracco@gmail.com (P.B.); rita.mattiello@ufrgs.br (R.M.); 2Postgraduate Program in Food, Nutrition and Health, School of Medicine, Universidade Federal do Rio Grande do Sul, Porto Alegre 90035-003, Rio Grande do Sul, Brazil

**Keywords:** gestational diabetes, gestational weight gain, early diagnosis of GDM

## Abstract

**Background/Objectives**: While clinical guidelines recommend screening and treatment for gestational diabetes mellitus (GDM) between 24 and 28 weeks, the benefits of earlier diagnosis are emerging. The objective of this study was to evaluate whether the early diagnosis of GDM is associated with reduced excessive gestational weight gain (GWG). **Methods:** Cohort study that analyzed 4694 pregnant women diagnosed with GDM attending high-risk prenatal care services within the Brazilian Unified Health System in six Brazilian capitals. GWG was classified according to Brazilian-specific pregnancy recommendations. ANCOVA tests were used to compare mean differences in total GWG across the timing of diagnosis. The timing of GDM diagnosis and excessive GWG was further evaluated using linear and logistic regression analysis. **Results:** Among the 4694 women with GDM (mean age 31.7 ± 6.3 years; mean pre-pregnancy BMI 30.4 ± 6.5 kg/m^2^, with 47.6% classified with obesity), those diagnosed in the first trimester (*n* = 1315) gained 2.29 kg less (95% CI: −2.87 to −1.71 kg) total GWG compared to the third trimester, adjusting for risk factors including pregestational weight. First-trimester GDM diagnosis was associated with 22% lower odds of experiencing excessive GWG (Odds Ratio [OR] = 0.78; 95% CI: 0.72–0.86), compared to the third trimester. Diagnoses before 20 weeks and before 24 weeks had 18% (OR = 0.82; 95% CI: 0.77–0.88) and 19% (OR = 0.81; 95% CI: 0.76–0.87) lower odds of excessive GWG. **Conclusions:** Early diagnosis of GDM, particularly during the first trimester, is associated with reduced GWG. Integrating earlier GDM screening into routine prenatal care could mitigate excessive GWG.

## 1. Introduction

Gestational diabetes mellitus (GDM) represents a significant global health challenge, currently affecting at least 14% of pregnancies worldwide [1]. Its escalating prevalence, closely tied to the global obesity epidemic and varying diagnostic practices, underscores the urgent need for effective management strategies [2]. Beyond its high incidence, GDM poses considerable risks, increasing both short- and long-term pregnancy complications for mother and offspring and is a major contributor to postpartum type 2 diabetes, accounting for up to 31% of cases [3].

Excessive gestational weight gain (GWG) is a risk factor for GDM and, in women with GDM, a risk factor for adverse outcomes, including hypertensive disorders of pregnancy, cesarean delivery, and fetal complications such as large for gestational age (LGA) and macrosomia [4,5]. Optimal management of GDM and excessive GWG, primarily through lifestyle interventions, is paramount to mitigate these risks and improve long-term outcomes.

Although it is recommended that GDM screening and subsequent treatment routinely occur between 24 and 28 weeks of gestation, early-onset GDM—diagnosed before 20 weeks—has gained increasing attention [6]. The prevalence of GDM diagnosed before 20 weeks of gestation, regardless of the diagnostic criteria used, ranges from 0.7% to 36.8% across countries [7]. Studies suggest that the risk of certain pregnancy complications is reduced by treating GDM before 20 weeks of gestation [8], with additional benefits including reductions in maternal and neonatal complications, overall costs, and improvements in Apgar scores, quality of life, and initiation of breastfeeding [9].

Despite the suggested benefits of early GDM detection and the critical importance of GWG management, the direct relationship between early GDM diagnosis and its impact on maternal weight gain trajectories remains poorly explored. Understanding this relationship is vital for refining clinical guidelines and optimizing the timing and effectiveness of interventions in GDM care. Therefore, the present study aims to evaluate whether the early diagnosis of GDM is associated with a lower incidence of excessive GWG.

## 2. Materials and Methods

### 2.1. Study Population and Design

The multicenter LINDA-Brasil cohort recruited pregnant women diagnosed with GDM from high-risk prenatal care services within the Brazilian Unified Health System (SUS) in six Brazilian cities: Pelotas (RS), Porto Alegre (RS), Curitiba (PR), Fortaleza (CE), Rio de Janeiro (RJ), and São Paulo (SP), between January 2014 and July 2019 [10].

We included women aged 18 years or older, without a prior diagnosis of type 2 diabetes, who were diagnosed with GDM according to criteria adopted at the time of their diagnosis. For inclusion, participants were required to be at least 32 weeks pregnant and onwards. Women were excluded if they had multiple pregnancies, a gestational age at delivery exceeding 44 weeks, missing pre-pregnancy Body Mass Index (BMI) data, or fewer than one recorded weight measurement during pregnancy. The final sample included 4694 women (Figure 1).

### 2.2. Data Collection

Data were collected by trained and certified interviewers using a standardized questionnaire. The questionnaire included demographic and socioeconomic variables such as the research center Brazilian region (Northeast, Southeast, and South), age (years), self-reported skin color (white and non-white), and education level (in years, categorized as <8, 8 to 12, ≥13). Mother and children clinical information extracted from interviews and prenatal care booklets included the parity (0, 1 to 2 and ≥3), pre-pregnancy BMI (<25, 25 to 30, ≥30), smoking status before pregnancy (no, yes), gestational age at GDM diagnosis (weeks), insulin use during pregnancy (no, yes), total gestational weight gain (kg), mode of delivery (vaginal, cesarean). Information on neonates was birth weight (g), preterm birth (GA < 37 weeks), extreme preterm birth (<34 weeks), low birth weight (<2500 g), and macrosomia (>4000 g).

### 2.3. Pre-Pregnancy Body Mass Index (BMI)

Pre-pregnancy BMI was calculated using the self-reported pre-pregnancy weight collected during the interview. Height was preferentially obtained from medical records; when unavailable, self-reported height was used. In cases where self-reported pre-pregnancy weight was implausible or missing, the first recorded weight measurement obtained up to the 14th week of gestation was utilized for BMI calculation. Pre-pregnancy nutritional status was classified according to World Health Organization (2000) criteria: normal (<25 kg/m^2^), overweight (≥25 and <30 kg/m^2^), or obesity (≥30 kg/m^2^) [11]. Underweight and normal weight categories were grouped due to the small number of women in the first and the similarity between the Brazilian growth charts for these two BMI classes [12].

### 2.4. Gestational Diabetes Mellitus Diagnostic Criteria

GDM diagnosis was confirmed through documentation in the women’s medical records (e.g., prenatal care booklet). The women included in our study were followed within the Brazilian Unified Health System (Sistema Único de Saúde—SUS), which recommends universal screening for gestational diabetes mellitus (GDM) with an OGTT between the 24th and 28th weeks of gestation. This diagnosis had been previously established by a prenatal care physician following the diagnostic criteria valid at the time of diagnosis, including guidelines from the World Health Organization [13], from the National Institutes of Health Conference [14], the International Association of Diabetes and Pregnancy Study Groups [15], and American Diabetes Association [16]. Only women with a documented GDM diagnosis and no prior history of pregestational diabetes were included in the study. To investigate the timing of gestational diabetes mellitus (GDM) diagnosis regarding excessive gestational weight gain, we categorized the diagnoses into three groups: first trimester (up to and including 13 weeks of gestation), second trimester (14 to 28 weeks), and third trimester (beyond 28 weeks). Additionally, we conducted separate analyses comparing women diagnosed before 20 weeks and before 24 weeks of gestation with those diagnosed later, to explore potential associations with excessive weight gain. Early GDM diagnosis was defined as occurring in the first trimester or before 20 weeks of gestation.

### 2.5. Gestational Age (GA)

GA at delivery was estimated primarily by comparing the first ultrasound scan with the date of the last menstrual period (LMP). The preferred method for GA estimation was an ultrasound performed before the 20th week of pregnancy. If the ultrasound was performed after 20 weeks (i.e., a late ultrasound), we examined whether the difference between the LMP and ultrasound estimate was one week or less. In such cases, the ultrasound was used for GA calculation. If the difference ranged from one to two weeks, the average between the LMP and the late ultrasound was used. When the difference between the LMP and late ultrasound ranged from 14 to 30 days, both sources were reviewed on a case-by-case basis, and the more reliable information was used. If the difference exceeded one month, GA was calculated as the average of all ultrasound estimates recorded in the prenatal care booklet.

### 2.6. Total Gestational Weight Gain (GWG) Calculation and Classification

GWG was calculated as the difference between the last recorded weight during pregnancy and the pre-pregnancy weight.

The last recorded prenatal weight was preferentially obtained from the participant’s prenatal care booklet. In instances where this data was unavailable or unrecorded in the booklet, the last weight reported by the participant during the immediate postpartum interview was utilized. To classify total GWG as excessive, we applied the Brazilian recommendations for gestational weight gain, which are based on pre-pregnancy Body Mass Index (BMI) [12,17,18].

Excessive total weight gain was defined as follows:

For women with BMI < 18.5 kg/m^2^: >12.2 kg;

For women with BMI ≥ 18.5 and <25 kg/m^2^: >12.0 kg;

For women with BMI ≥ 25 and <30 kg/m^2^: >9.0 kg;

For women with BMI ≥ 30 kg/m^2^: >7.2 kg.

Cumulative weight gain by gestational age was calculated by subtracting pre-pregnancy weight from each recorded prenatal weight measurement. To estimate weight gain during the first trimester, weight measurements from the 10th week onward were considered. For the second trimester, the most recent measurement between the 23rd and 27th weeks of gestation was used. To calculate the final cumulative weight gain, the last weight measured between the 37th and 42nd gestational weeks was considered. Cumulative weight gain by gestational week was compared to the 2023 established Brazilian GWG reference charts [12].

### 2.7. Postpartum Follow-Up

Postpartum follow-up was conducted through telephone interviews. During these calls, information was collected on delivery (date of delivery, last weight measured while still pregnant), newborn data (health status and birth weight), and the use of medications for glycemic control during pregnancy, including the last weight measurement during gestation.

### 2.8. Statistical Analysis

Data cleaning procedures were performed. The analysis of outliers considered maternal variables such as pre-pregnancy weight, total gestational weight gain (GWG), and pre-pregnancy Body Mass Index (BMI) for each trimester. Outliers were identified using Z-scores from the sample distribution with a cutoff point of ±4 Z-scores for potential exclusion [19].

Means and standard deviations were used for continuous variables, while absolute and relative frequencies were used for categorical variables.

To classify GWG values from the LINDA-Brasil study with the new Brazilian reference charts [12], we considered the number of weight observations recorded during prenatal care, their respective gestational weeks, and their differences from pre-pregnancy weight, stratified by pre-pregnancy BMI categories. The GWG trajectories observed in this study were plotted onto the reference charts for each BMI category and evaluated according to the 10th, 25th, 50th, 75th, and 90th percentiles. Adequate weight gain was defined as GWG falling within the shaded areas of the curves, which correspond to the following percentile intervals: p10–p34 for women with normal pre-pregnancy BMI, p18–p27 for women with overweight, and p27–p38 for women with obesity [12]. ANCOVA with post hoc Bonferroni adjustment was used to assess significant differences in mean total GWG according to the trimester of GDM diagnosis. Models were adjusted for maternal age, region, skin color, pre-pregnancy BMI, smoking before and during pregnancy, use of medication for gestational diabetes, and recommendation for physical activity during pregnancy.

To evaluate the association between excessive GWG and early or late GDM diagnosis, linear and logistic regression analyses were used. The models considered the trimester of GDM diagnosis, and, for the logistic regression outcome, GWG was categorized as excessive or non-excessive. Excessive weight gain was considered if the women exceeded the weight gain recommendation at either the second or the third trimester. The models were adjusted for covariates considered potential confounding factors, including maternal age, region, skin color, pre-pregnancy BMI, smoking before and during pregnancy, use of medication for gestational diabetes, and recommendation for physical activity during pregnancy. The outcome was assessed for normality. We did graphical residual analysis for all models to check the assumptions of homoscedasticity, normality, and independence of residuals. For logistic regression, we used the simulated half-normal envelope plot. All statistical analyses were performed using R software (version 4.4.3).

The project was approved by the ethics committees of the Hospital de Clínicas de Porto Alegre (HCPA), the Federal University of Pelotas (UFPEL), and the Center for Diabetes and Hypertension Studies in Fortaleza (CE). All participants signed a written informed consent form.

## 3. Results

Table 1 presents the characteristics of the study sample. The mean age of participants was 31.7 years (SD ± 6.3). Most women were (64.5%) from the South Brazilian Region, (52.5%) between 30 and 39 years old, followed by those aged 18 to 29 (36.3%) and ≥40 years (11.3%). Most (53.8%) reported being non-White, with 8 to 12 years of schooling (54.4%) and having 1 to 2 children. The mean pre-pregnancy BMI was 30.4 kg/m^2^ (SD ± 6.5), with 47.6% of the women being classified as having obesity (BMI ≥ 30). Pre-pregnancy smoking was reported by 21.5% of the women. The mean gestational age at GDM diagnosis was 19.9 ± 8.2 weeks, gestational weight gain and age at delivery were 9.1 ± 7.7 and 38.3 ± 2.1 weeks, respectively. Cesarean delivery occurred in 62.8% of the sample, with a mean birth weight of 3295.2 ± 712.9 g. The rates of insulin use during pregnancy, preterm birth, low birth weight, and macrosomia were 41.1%,10.7%, 6.9%, and 7.6%, respectively.

Figure 2 shows that participants from the LINDA-Brasil study exhibited GWG trajectories (cumulative maternal weight gain curves in kg) that exceeded the Brazilian reference curves, regardless of the trimester in which GDM was diagnosed and stratified by pre-pregnancy BMI. Across all pre-pregnancy BMI categories, accelerated weight gain was observed early in pregnancy. Women diagnosed with GDM in the first trimester reached the 25th and 50th percentiles of GWG earlier and maintained within those ranges throughout pregnancy, irrespective of pre-pregnancy BMI. Conversely, women diagnosed in the second and third trimesters presented GWG trajectories near the 75th percentile, with deceleration in weight gain occurring later.

Table 2 presents the adjusted mean total GWG according to the trimester and weeks of GDM diagnosis. After descriptive and statistical analysis using the Shapiro-Wilks test, the total GWG outcome did not deviate significantly from normality. The ANCOVA model was adjusted for maternal age, Brazilian region, skin color, smoking before and during pregnancy, use of GDM medication, pre-pregnancy BMI, and physical activity recommendations during pregnancy. The mean GWG differed significantly by trimester, particularly between the first and third, and by weeks of diagnosis (20 or 24 wk). Women diagnosed in the first trimester had a mean GWG of 8.37 kg (95% CI: 7.88–8.87), those diagnosed in the second trimester gained 9.43 kg (95% CI: 9.00–9.86), and the highest GWG was observed among those diagnosed in the third trimester, at 10.66 kg (95% CI: 10.16–11.17). Women diagnosed before week 20 had a mean GWG of 8.56 kg (95% CI 8.12–8.99), and 8.81 kg (95% CI 8.40–9.22) when the diagnosis was before week 24.

Table 3 presents the adjusted association of total gestational weight gain means and either the trimester of GDM diagnosis or the week of GDM diagnosis. GDM diagnosis in the first trimester was associated with a decrease in GWG of −2.29 kg (95% CI: −2.87; −1.71) compared with a diagnosis in the 3rd trimester. Second-trimester diagnosis was associated with a decrease in GWG of −1.23 kg (95% CI: −1.75; −0.72) compared to the third trimester. A diagnosis before 20 and 24 weeks was associated with a decrease in GWG of −1.58 (95% CI: −2.01; −1.16) and −1.44 (95% CI: −1.86; −1.02), when compared with after 20 weeks and after 24 weeks, respectively.

Table 4 shows the odds ratios for excessive GWG according to the trimester of GDM diagnosis, adjusted for age, study center, skin color, education level, pre-pregnancy smoking, region, pre-pregnancy BMI, smoking during pregnancy, use of GDM medication, and physical activity recommendation. Early GDM diagnosis, particularly in the first trimester, was associated with excessive GWG. Women diagnosed in the first trimester had 22% lower odds of excessive GWG compared to those diagnosed in the third trimester (OR = 0.78, 95% CI: 0.72–0.86). Women with a diagnosis before 20 weeks and before 24 weeks had 18% (OR = 0.82, 95% CI: 0.77–0.88) and 19% (OR = 0.81, 95% CI: 0.76–0.87) lower odds of excessive GWG.

## 4. Discussion

This large-scale cohort study provides evidence that the early diagnosis of gestational diabetes mellitus, particularly in the first trimester and before 20 weeks of pregnancy, is significantly associated with reduced total gestational weight gain and a lower incidence of excessive GWG. Our findings reveal that women diagnosed with GDM in the first trimester had 22% lower odds of excessive GWG compared to those diagnosed in the third trimester. While its magnitude was modest, this effect size holds clinical relevance, particularly when considering the extensive complications that excessive gestational weight gain can lead to for both offspring and mothers. Crucially, these results underscore the critical role of timely GDM identification in facilitating earlier and potential for earlier implementation of nutritional and lifestyle interventions, which are cornerstones for preventing and managing complications associated with GDM and excessive GWG.

Our findings, showing a 2.29 kg reduction in total weight gain for those diagnosed in the first trimester compared to the third, align with previous studies such as Hillier et al. (2020) and others that demonstrate similar benefits of early GDM detection on maternal weight gain trajectories [20,21,22]. These consistent observations suggest a direct link between the timing of diagnosis and the efficacy of subsequent management strategies. While clinical guidelines typically recommend GDM screening between 24 and 28 weeks, the American Diabetes Association already advocates for early pregnancy screening in high-risk women to identify and treat unrecognized diabetes [23]. Our study provides robust real-world evidence supporting this call.

Indeed, nutritional interventions and lifestyle modifications constitute the cornerstone of effective GDM and obesity management. When GDM is diagnosed early, pregnant women receive guidance on healthy eating patterns, appropriate portion control, and regular physical activity significantly sooner in their pregnancy. According to the guidelines of the Brazilian Diabetes Society, daily self-monitoring of capillary blood glucose is recommended to begin immediately after the diagnosis of GDM, with earlier implementation being associated with greater effectiveness, and should be continued until delivery. Intensive self-monitoring has been associated with lower rates of macrosomia, cesarean delivery, shoulder dystocia, stillbirth, and neonatal intensive care unit (NICU) admissions [24]. Our findings reinforce the potential utility of earlier GDM screening in low- and middle-income countries (LMICs), where primary health care systems, such as Brazil’s community-based model, can facilitate early detection and management through territorial coverage and the active role of community health agents in identifying pregnancies at an early stage. In such settings, primary care units are equipped to initiate timely, non-pharmacological interventions and, when necessary, pharmacological treatment, with referral to high-risk prenatal care when glycemic targets are not met. These findings align with recent global recommendations for a life-course approach to diabetes prevention and management, as outlined by Simmons et al. (2024), and underscore how early GDM diagnosis can be both feasible and impactful in LMICs when supported by existing public health infrastructure [25]. This early awareness and proactive management empower women to adopt crucial behavioral changes that effectively modulate weight gain throughout gestation [26].

Our data strikingly illustrate the following: women diagnosed in the first trimester, despite initial accelerated weight gain, maintained GWG trajectories closer to the recommended percentiles. This contrasts sharply with those diagnosed later, whose weight gain deceleration occurred much later. This aligns with prior research demonstrating the benefits of structured lifestyle interventions in GDM management, including improved glycemic control and reduced adverse outcomes [27].

Restricting GWG after a GDM diagnosis is particularly advantageous, especially in women who have experienced excessive GWG in the first half of pregnancy, a characteristic prevalent in our evaluated population. The physiological link between hyperglycemia and increased neonatal weight, possibly due to upregulated glucose transporter-1 (GLUT1) and elevated circulating lipid levels in GDM placentas, further underscores the importance of early and effective glycemic and weight management [28,29]. Our findings add to existing evidence that less GWG mediates, at least in part, the neonatal benefits observed in major trials of early GDM treatment [8].

A key strength of our study is its large sample size (*n* = 4694) drawn from a diverse, high-risk population receiving care within a public health system, enhancing the generalizability of our findings to similar settings. The use of pre-pregnancy BMI to stratify GWG outcomes, following Brazilian recommendations [18], adds further robustness to our analysis. Moreover, our comprehensive adjustment for a wide range of potential confounders strengthens the validity of the observed associations. However, some limitations should be acknowledged.

As an observational study, we cannot establish causality between early diagnosis and reduced GWG; rather, we highlight a strong association with a cohort study. The reliance on self-reported data for pre-pregnancy weight and some clinical information introduces potential for recall bias. However, we based our approach on evidence from Carrilho et al. (2020), which demonstrated a high level of agreement between self-reported pre-pregnancy weight and measured weight in the first trimester among Brazilian women [30]. Additionally, while we infer that lifestyle interventions are the mediating factor, our study did not directly quantify the intensity, content, or adherence to specific dietary and physical activity interventions initiated following GDM diagnosis. This limits our ability to determine which specific aspects of the intervention were most effective. Moreover, although we adjusted for pre-pregnancy BMI and medication use, data on dietary habits and physical activity were not available for the entire study sample, preventing us from including these potential confounders in the adjusted regression models. We also acknowledge that GWG is a surrogate marker for critical maternal and neonatal outcomes, such as neonatal macrosomia and preeclampsia. Finally, the Brazilian GWG reference charts used in this study have not yet been validated in other populations, which may limit the generalizability of our findings to settings with different BMI distributions and maternal characteristics.

Our findings support the potential benefits of earlier screening for gestational diabetes, particularly relevant for health care settings where early screening is not yet routine. Accordingly, these results carry important implications for clinical practice in regions like Brazil and other low- and middle-income countries. Implementing earlier screening protocols in these contexts requires careful consideration of the balance between their clinical utility, feasibility, and the potential costs or resource limitations involved.

## 5. Conclusions

In conclusion, early GDM diagnosis, particularly in the first trimester, is significantly associated with reduced total gestational weight gain and a lower incidence of excessive GWG. This highlights the critical role of timely GDM identification in promoting better maternal weight trajectories, a crucial determinant of both maternal and offspring health outcomes. These results help advocate a paradigm shift towards earlier GDM diagnosis, enabling timely and effective lifestyle and nutritional interventions to optimize maternal and fetal health.

## Figures and Tables

**Figure 1 nutrients-17-02600-f001:**
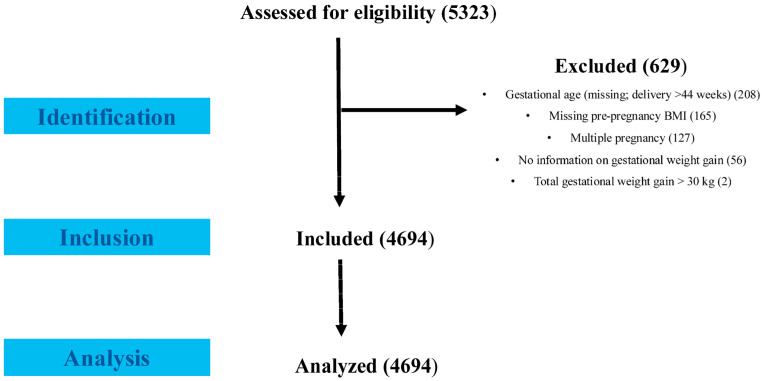
Flowchart of the sample showing the number of participants and exclusions applied in the analyses.

**Figure 2 nutrients-17-02600-f002:**
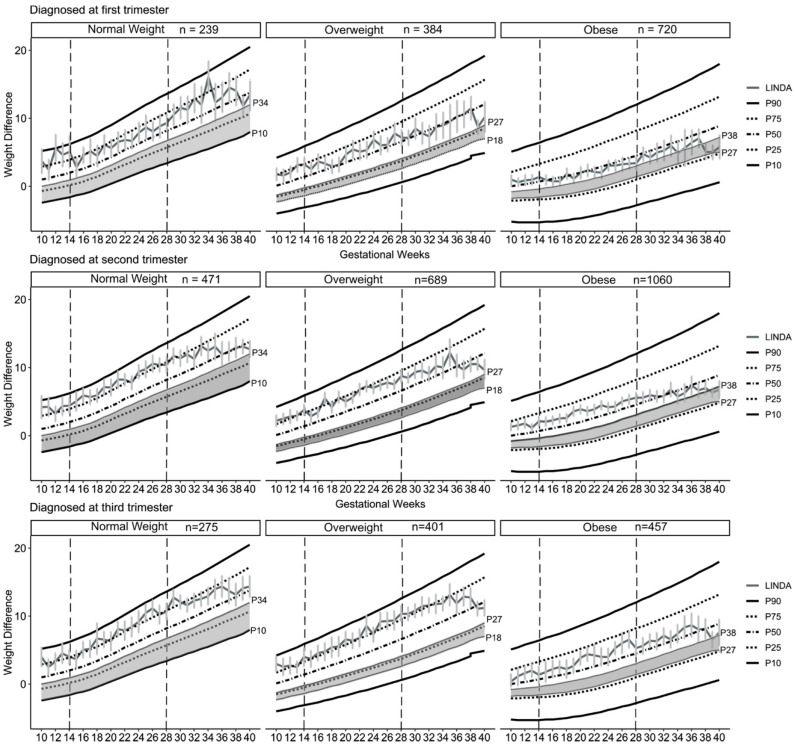
Cumulative maternal weight gain curves (in kg) across gestational weeks, according to the trimester of gestational diabetes diagnosis and stratified by pre-pregnancy BMI. The gray-shaded area represents the Brazilian gestational weight gain recommendations.

**Table 1 nutrients-17-02600-t001:** Sample characteristics overall and according to the trimester of gestational diabetes diagnosis.

Sample Characteristics	Total Sample *n* = 4694	Timing of Gestational Diabetes Diagnosis		
		1st Trimester *n* = 1342	2nd Trimester *n* = 2220	3rd Trimester *n* = 1132
Brazilian Region/Center, *n* (%)				
Northeast	1120 (23.9%)	175 (13.0%)	575 (25.9%)	370 (32.7%)
Southeast	539 (11.5%)	178 (13.3%)	199 (9.0%)	162 (14.3%)
South	3035 (64.6%)	989 (73.7%)	1446 (65.1%)	600 (53.0%)
Age (years), mean ± SD	31.7 ± 6.3	31.1 ± 6.3	32.0 ± 6.2	31.7 ± 6.4
18 to 29, *n* (%)	1679 (36.2)	518 (39.1)	748 (34.1)	413 (37.2)
30 to 39, *n* (%)	2430 (52.5)	686 (51.8)	1178 (53.7)	566 (50.9)
≥40 *n* (%)	522 (11.3)	121 (9.1)	269 (12.2)	132 (11.9)
Skin color, *n* (%)				
White	2167 (46.2)	689 (51.3)	1010 (45.5)	468 (41.3)
Non-White	2527 (53.8)	653 (48.7)	1210 (54.5)	664 (58.7)
Education (years), mean ± SD				
<8, *n* (%)	1365 (29.1)	389 (29.0)	656 (29.5)	320 (28.3)
8 to 12, *n* (%)	2556 (54.4)	725 (54.0)	1218 (54.9)	613 (54.1)
≥13, *n* (%)	773 (16.5)	228 (17.0)	346 (15.6)	199 (17.6)
Parity				
0, *n* (%)	1478 (31.5)	444 (33.1)	666 (30.0)	368 (32.5)
1 to 2, *n* (%)	2535 (54.0)	716 (53.3)	1212 (54.6)	607 (53.6)
≥3, *n* (%)	680 (14.5)	182 (13.6)	341 (15.4)	157 (13.9)
Pre-pregnancy BMI (kg/m^2^), mean ± SD	30.4 ± 6.5	31.2 ± 6.8	30.3 ± 6.5	29.4 ± 6.2
<25, *n* (%)	984 (21.0)	239 (17.8)	471 (21.2)	274 (24.2)
25 to 30, *n* (%)	1474 (31.4)	384 (28.6)	689 (31.0)	401 (35.4)
≥30, *n* (%)	2236 (47.6)	719 (53.6)	1060 (47.8)	457 (40.4)
Smoking before pregnancy				
No, *n* (%)	3685 (78.5)	1033 (77.0)	1763 (79.5)	889 (78.5)
Yes, *n* (%)	1007 (21.5)	308 (23.0)	456 (20.5)	243 (21.5)
GA at GDM diagnosis (weeks), mean ± SD	19.9 ± 8.2	8.9 ± 2.5	21.7 ± 3.7	29.3 ± 2.3
Insulin use during pregnancy, *n* (%)				
No	2766 (58.9)	705 (52.5)	1270 (57.2)	791 (69.9)
Yes	1928 (41.1)	637 (47.5)	950 (42.8)	341 (30.1)
Total gestational weight gain (kg) mean ± SD	9.1 ± 7.7	8.0 ± 7.8	9.0 ± 7.5	10.5 ± 7.5
Gestational age at delivery (weeks) mean ± SD	38.3 ± 2.1	38.2 ± 1.9	38.2 ± 2.2	38.5 ± 1.9
Mode of delivery				
Vaginal, *n* (%)	1654 (37.1)	484 (38.0)	770 (36.7)	400 (37.0)
Cesarean, *n* (%)	2799 (62.9)	790 (62.0)	1329 (63.3)	680 (63.0)
Birth weight (g) mean ± SD	3295.2 ± 712.9	3297.5 ± 1010.8	3275.8 ± 566.9	3330.4 ± 508.5
Preterm birth (GA < 37 weeks), *n* (%)	504 (10.7)	141 (10.5)	258 (11.7)	105 (9.3)
Extreme preterm birth (<34 weeks), *n* (%)	94 (2.0)	28 (2.1)	58 (2.6)	8 (0.7)
Low birth weight (<2500 g), *n* (%)	301 (6.9)	85 (6.7)	162 (7.8)	54 (5.1)
Macrosomia (>4000 g), *n* (%)	331 (7.6)	86 (6.83)	155 (7.5)	90 (8.5)

SD = Standard Deviation; BMI = Body Mass Index; GA = Gestational Age. Data are *n* (%) or ± SD.

**Table 2 nutrients-17-02600-t002:** Adjusted means of total gestational weight gain by trimester and by weeks of gestational diabetes mellitus (GDM) diagnosis.

GDM Diagnosis	*n* (%)	Total Gestational Weight Gain (kg)
		Adjusted Mean * (95% CI)
Total Sample	4597 (100)	
Diagnosis in the 1st trimester	1315 (28.6%)	8.37 (7.88 to 8.87) ^a^
Diagnosis in the 2nd trimester	2178 (47.4%)	9.43 (9.00 to 9.86) ^b^
Diagnosis in the 3rd trimester	1104 (24.0%)	10.66 (10.16 to 11.17) ^c^
Diagnosis before 20 weeks	1925 (41.9%)	8.56 (8.12 to 8.99) ^a^
Diagnosis after 20 weeks	2672 (58.1%)	10.14 (9.73 to 10.55) ^b^
Diagnosis before 24 weeks	2563 (55.8%)	8.81 (8.40 to 9.22) ^a^
Diagnosis after 24 weeks	2034 (44.2%)	10.25 (9.82 to 10.68) ^b^

* ANCOVA; Models adjusted for: maternal age, region, skin color, pre-pregnancy BMI, smoking before and during pregnancy, use of medication for gestational diabetes, and recommendation for physical activity during pregnancy. Means followed by the same letter do not differ significantly from each other (post hoc Bonferroni adjustment).

**Table 3 nutrients-17-02600-t003:** Association between timing of gestational diabetes diagnosis and total gestational weight gain adjusted for confounders.

	*n* (%)	Total Gestational Weight Gain Adjusted Model *
		β 95% CI
Total Sample	4597 (100)	
Diagnosis in the 1st trimester	1315 (28.6%)	−2.29 (−2.87 to −1.71)
Diagnosis in the 2nd trimester	2178 (47.4%)	−1.23 (−1.75 to −0.72)
Diagnosis in the 3rd trimester	1104 (24.0%)	ref
Diagnosis before 20 weeks	1925 (41.9%)	−1.58 (−2.01 to −1.16)
Diagnosis after 20 weeks	2672 (58.1%)	ref
Diagnosis before 24 weeks	2563 (55.8%)	−1.44 (−1.86 to −1.02)
Diagnosis after 24 weeks	2034 (44.2%)	ref

* Linear regression models adjusted for: maternal age, region, skin color, pre-pregnancy BMI, smoking before and during pregnancy, use of medication for gestational diabetes, and recommendation for physical activity during pregnancy.

**Table 4 nutrients-17-02600-t004:** Association between timing of gestational diabetes diagnosis and excessive gestational weight gain according to the new Brazilian recommendations.

	*n* (%)	Excessive Gestational Weight Gain Adjusted Model *
		OR 95% CI
Total Sample	3942 (100)	
Diagnosis in the 1st trimester	1178 (29.9%)	0.78 (0.72 to 0.86)
Diagnosis in the 2nd trimester	1868 (47.4%)	0.90 (0.83 to 0.97)
Diagnosis in the 3rd trimester	896 (22.7%)	ref
Diagnosis before 20 weeks	1705 (43.3%)	0.82 (0.77 to 0.88)
Diagnosis after 20 weeks	2237 (56.7%)	ref
Diagnosis before 24 weeks	2246 (57.0%)	0.81 (0.76 to 0.87)
Diagnosis after 24 weeks	1696 (43.0%)	ref

* Logistic Regression models adjusted for: maternal age, region, ethnicity, pre-pregnancy BMI, smoking before and during pregnancy, use of medication for gestational diabetes, and recommendation for physical activity during pregnancy.

## Data Availability

The data analyzed in this study are subject to the following licenses/restrictions: no restriction. Requests to access these datasets should be directed to midrehmer@hcpa.edu.br.

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
