# Peer review of "Early Gestational Diabetes Mellitus Diagnosis: A Strategy for Mitigating Excessive Maternal Weight Gain—LINDA-Brasil Study"

_nutrients, 2025, doi:10.3390/nu17162600_

Round 1

Reviewer 1 Report

Comments and Suggestions for Authors

The article "Early Gestational Diabetes Mellitus Diagnosis: A Strategy for Mitigating Excessive Maternal Weight Gain" by Letícia Ribeiro Pavão da Silveira et al. offers valuable contributions, but the following points require clarification or elaboration:

  1. Were all participants uniformly screened (e.g., OGTT) across trimesters? If screening was risk-factor-based (e.g., obesity), early-diagnosed women may represent a distinct subgroup, introducing selection bias.
  2. While pre-pregnancy BMI was adjusted for, other potential confounders (e.g., dietary habits, physical activity, medication use) that influence gestational weight gain (GWG) were not addressed.
  3. The cohort design cannot establish causality. Consider reframing conclusions (e.g., "early diagnosis is associated withreduced GWG" rather than "leads to").
  4. Does reduced GWG correlate with improved maternal/neonatal outcomes (e.g., lower rates of C-sections, macrosomia)? Without such data, the clinical implications remain unclear.
  5. The terms "first-trimester diagnosis," "<20 weeks," and "<24 weeks" are used interchangeably. Clarify diagnostic criteria, as standard GDM screening rarely occurs before 24 weeks. Were diagnoses protocol-driven or opportunistic (e.g., due to symptoms/high-risk status)? This impacts generalizability.
  6. GWG is a surrogate marker. Provide evidence linking it to hard endpoints (e.g., neonatal macrosomia, preeclampsia) to justify early screening’s clinical utility.
  7. The study employs Brazilian GWG thresholds. Would findings generalize to populations with divergent BMI distributions (e.g., Asian vs. European)?
  8. The reported OR for first-trimester diagnosis (0.78, 95% CI: 0.72–0.86) was statistically significant but modest. Discuss whether this effect size is clinically meaningful in practice.
Comments on the Quality of English Language

The manuscript requires thorough English language editing to improve clarity, grammar, and overall readability

Author Response

Response letter

“Early Gestational Diabetes Mellitus Diagnosis: A Strategy for Mitigating Excessive Maternal Weight Gain – LINDA-Brasil"

Manuscript Number: nutrients-3788579

Dear Dr. Immanuel and Dr. Simmons,

Editors-in-Chief

Nutrients

We appreciate the comments and suggestions made by the reviewers and the opportunity to revise the manuscript. All changes made to the manuscript are highlighted in red font, and point-by-point responses to the reviewers’ comments are provided below.

Reviewers' comments:

Comments and Suggestions for Authors

The article "Early Gestational Diabetes Mellitus Diagnosis: A Strategy for Mitigating Excessive Maternal Weight Gain" by Letícia Ribeiro Pavão da Silveira et al. offers valuable contributions, but the following points require clarification or elaboration:

1. Were all participants uniformly screened (e.g., OGTT) across trimesters? If screening was risk-factor-based (e.g., obesity), early-diagnosed women may represent a distinct subgroup, introducing selection bias.

Response: Thank you for your comment and for raising this important point. The women included in our study were followed within the Brazilian Unified Health System (Sistema Único de Saúde – SUS), which recommends universal screening for gestational diabetes mellitus (GDM) with an OGTT between the 24th and 28th weeks of gestation. The test was administered in a real-world clinical setting and early screening could have been performed based on risk factors. We made adjustments for this as discussed below.

Lines: 115-118

2. While pre-pregnancy BMI was adjusted for, other potential confounders (e.g., dietary habits, physical activity, medication use) that influence gestational weight gain (GWG) were not addressed.

Response: Thank you for this valuable comment. Regarding medication use, we confirm that this variable was adjusted for in our analytical models. Unfortunately, data on dietary habits and physical activity were collected only for a small subsample. We emphasized this limitation in the revised manuscript.

Lines: 377-380

3. The cohort design cannot establish causality. Consider reframing conclusions (e.g., "early diagnosis is associated with reduced GWG" rather than "leads to").

Response: Thank you for the suggestion. We replaced 'leads to' with “is associated with”.  

Lines: 20, 34-35,  395

4. Does reduced GWG correlate with improved maternal/neonatal outcomes (e.g., lower rates of C-sections, macrosomia)? Without such data, the clinical implications remain unclear.

Response: Thank you for this important observation. We acknowledge that analyses exploring maternal and neonatal outcomes, such as cesarean section or macrosomia, would provide additional clinical insights. However, our primary objective was to investigate excessive gestational weight gain as the outcome and examine how the timing of gestational diabetes diagnosis may influence weight gain outside recommended guidelines.

  1. The terms "first-trimester diagnosis," "<20 weeks," and "<24 weeks" are used interchangeably. Clarify diagnostic criteria, as standard GDM screening rarely occurs before 24 weeks. Were diagnoses protocol-driven or opportunistic (e.g., due to symptoms/high-risk status)? This impacts generalizability.

Response: Thank you. We acknowledge the inconsistency in the terminology and have revised the manuscript to use the term "diagnosis before 20 weeks" consistently. This cutoff was chosen based on recent discussions in the literature questioning the potential benefits of diagnosing gestational diabetes before the standard 24–28-week window recommended by the WHO. This cutoff was specifically chosen in light of recent literature – notably Sweeting et al. in The Lancet, highlighting that 30–70% of gestational diabetes cases are diagnosed before 20 weeks and emphasizing the potential distinct outcomes associated with early‑pregnancy hyperglycemia. Our analyses aimed to explore whether earlier diagnosis, before 20 weeks, was associated with a greater risk of excessive gestational weight gain. The diagnoses in our cohort were not protocol-driven by the study but occurred in real-world clinical practice within the public prenatal care system in Brazil. However, as discussed above, we cannot rule out early referral to high-risk pregnancy clinics associated with being at high-risk status and made adjustments for risk factors in analyses.

We added this explanation to the methods section:

“To investigate the timing of gestational diabetes mellitus (GDM) diagnosis regarding excessive gestational weight gain, we categorized the diagnoses into three groups: first trimester (up to and including 13 weeks of gestation), second trimester (14 to 28 weeks), and third trimester (beyond 28 weeks). Additionally, we conducted separate analyses comparing women diagnosed before 20 weeks and before 24 weeks of gestation with those diagnosed later, to explore potential associations with excessive weight gain. Early GDM diagnosis was defined as occurring in the first trimester or before 20 weeks of gestation.”

Lines: 124-130

6. GWG is a surrogate marker. Provide evidence linking it to hard endpoints (e.g., neonatal macrosomia, preeclampsia) to justify early screening’s clinical utility.

Response: We agree that GWG is a surrogate marker. Even though prior studies have established its connection to crucial hard endpoints (e.g., neonatal macrosomia, preeclampsia), we have included the inherent limitation of its surrogate status, specific to our study, in the discussion.

Lines: 380-382

7. The study employs Brazilian GWG thresholds. Would findings generalize to populations with divergent BMI distributions (e.g., Asian vs. European)?

Response: The new Brazilian curves have not yet been tested in other populations. We have included this in the study's limitations.

Lines: 382-385

8. The reported OR for first-trimester diagnosis (0.78, 95% CI: 0.72–0.86) was statistically significant but modest. Discuss whether this effect size is clinically meaningful in practice.

Response: While the odds ratio for first-trimester diagnosis (0.78, 95% CI: 0.72–0.86) achieved statistical significance, its magnitude was modest. Nevertheless, this effect size holds clinical relevance, particularly when considering the extensive complications that excessive gestational weight gain can lead to for both offspring and mothers. We added this information to the discussion.

Lines: 310-315

Comments on the Quality of English Language

The manuscript requires thorough English language editing to improve clarity, grammar, and overall readability.

Response: The manuscript has been subjected to thorough English language editing to enhance clarity, grammar, and overall readability.

Reviewer 2 Report

Comments and Suggestions for Authors

I read with interest the manuscript “Early Gestational Diabetes Mellitus Diagnosis: A Strategy for Mitigating Excessive Maternal Weight Gain”.

This is a well-structured and well-executed cohort study addressing an increasingly important issue in prenatal care: the potential benefits of early gestational diabetes mellitus (GDM) diagnosis in reducing excessive gestational weight gain (GWG). The manuscript is methodologically sound, clearly written, and clinically relevant. It provides robust real-world evidence from a large and diverse population sample in Brazil, adding weight to the call for earlier screening policies in high-risk groups.

Major concern

  1. While the authors acknowledge the observational nature of their study, several statements in the abstract and conclusion still imply causality (e.g., “early diagnosis reduces GWG”). It would be more scientifically accurate to frame this as an association rather than a cause-effect relationship.
  2. The manuscript infers that earlier diagnosis allows for earlier lifestyle interventions, yet no data is provided on whether and when participants received nutritional counseling, physical activity recommendations, or behavioral support. This limits the ability to confirm mechanisms. Even if not available, this limitation should be more clearly emphasized in the Discussion.
  3. The use of self-reported pre-pregnancy weight and, in some cases, first-trimester weight to estimate BMI may introduce misclassification bias. Although the authors mention this, further sensitivity analysis or subgroup analysis (e.g., excluding women with imputed weight) would strengthen the findings.
  4. The study convincingly supports the benefit of early diagnosis in mitigating excessive GWG. However, the authors should more explicitly discuss how these findings could or should influence screening practices, especially in settings where early screening is not yet routine. Consider highlighting the balance between clinical utility and cost/resource implications.

Minor comments:

  1. The use of ANCOVA, logistic, and linear regression is appropriate. However, details about model fit (e.g., R², assumptions checks) are missing and should be briefly discussed or noted.
  2. Ensure consistent use of units (e.g., kg, weeks) and clarify abbreviations (e.g., define “GWG” and “GDM” at first mention in abstract and body). Some table titles (e.g., Table 2) could benefit from clearer phrasing (e.g., specify if weights are adjusted means).

Author Response

Response letter

“Early Gestational Diabetes Mellitus Diagnosis: A Strategy for Mitigating Excessive Maternal Weight Gain – LINDA-Brasil"

Manuscript Number: nutrients-3788579

Dear Dr. Immanuel and Dr. Simmons,

Editors-in-Chief

Nutrients

We appreciate the comments and suggestions made by the reviewers and the opportunity to revise the manuscript. All changes made to the manuscript are highlighted in red font, and point-by-point responses to the reviewers’ comments are provided below.

Reviewers' comments:

Comments and Suggestions for Authors

I read with interest the manuscript “Early Gestational Diabetes Mellitus Diagnosis: A Strategy for Mitigating Excessive Maternal Weight Gain”.

This is a well-structured and well-executed cohort study addressing an increasingly important issue in prenatal care: the potential benefits of early gestational diabetes mellitus (GDM) diagnosis in reducing excessive gestational weight gain (GWG). The manuscript is methodologically sound, clearly written, and clinically relevant. It provides robust real-world evidence from a large and diverse population sample in Brazil, adding weight to the call for earlier screening policies in high-risk groups.

Major concern

  1. While the authors acknowledge the observational nature of their study, several statements in the abstract and conclusion still imply causality (e.g., “early diagnosis reduces GWG”). It would be more scientifically accurate to frame this as an association rather than a cause-effect relationship.

Response: We appreciate your suggestion. We are now framing our findings as an association rather than a cause-and-effect relationship.

Line: 34-35, 395

2. The manuscript infers that earlier diagnosis allows for earlier lifestyle interventions, yet no data is provided on whether and when participants received nutritional counseling, physical activity recommendations, or behavioral support. This limits the ability to confirm mechanisms. Even if not available, this limitation should be more clearly emphasized in the Discussion.

Response: We emphasize that early diagnosis can facilitate earlier lifestyle interventions, including offering more time for the intervention. We changed the statement regarding lifestyle information in the discussion accordingly.

Lines: 312-315

3. The use of self-reported pre-pregnancy weight and, in some cases, first-trimester weight to estimate BMI may introduce misclassification bias. Although the authors mention this, further sensitivity analysis or subgroup analysis (e.g., excluding women with imputed weight) would strengthen the findings.

Response: Thank you for this comment. We acknowledge the potential for misclassification when using self-reported pre-pregnancy weight or, in some cases, first-trimester weight to estimate BMI. However, we based our approach on evidence from Carrilho et al. (2020), which demonstrated a high level of agreement between self-reported pre-pregnancy weight and measured weight in the first trimester among Brazilian women (BMC Pregnancy and Childbirth, 2020; 20:734. https://doi.org/10.1186/s12884-020-03354-4). Given this strong concordance, we believe the likelihood of substantial bias being introduced under these circumstances is limited.

Lines: 371-373

4. The study convincingly supports the benefit of early diagnosis in mitigating excessive GWG. However, the authors should more explicitly discuss how these findings could or should influence screening practices, especially in settings where early screening is not yet routine. Consider highlighting the balance between clinical utility and cost/resource implications.

Response: We appreciate the reviewer’s comment. We agree that a more explicit discussion of the practical implications of our findings is essential. Accordingly, we have expanded the discussion to emphasize how our results support the potential benefits of earlier screening for gestational diabetes, particularly in healthcare settings where early screening is not routine. We also addressed the possible implications for clinical practice in Brazil and other low- and middle-income countries, considering the balance between clinical utility, feasibility, and the potential costs or resource limitations associated with implementing earlier screening protocols.

Lines: 328-344

Minor comments:

  1. The use of ANCOVA, logistic, and linear regression is appropriate. However, details about model fit (e.g., R², assumptions checks) are missing and should be briefly discussed or noted.

Response: Thank you. We included a more detailed description of the model fit in the article. We did graphical residual analysis for all models to check the assumptions of homocedasticity, normality, and independence of residuals. For logistic regression, we used the simulated half-normal envelope plot.

Lines: 205-208

2. Ensure consistent use of units (e.g., kg, weeks) and clarify abbreviations (e.g., define “GWG” and “GDM” at first mention in abstract and body). Some table titles (e.g., Table 2) could benefit from clearer phrasing (e.g., specify if weights are adjusted means).

Response: We have standardized units and clarified abbreviations. Furthermore, table titles have been reviewed and rephrased for improved clarity.

Round 2

Reviewer 1 Report

Comments and Suggestions for Authors

Accept in present form